# Transcriptome Analysis Reveals Cross-Talk between the Flagellar Transcriptional Hierarchy and Secretion System in *Plesiomonas shigelloides*

**DOI:** 10.3390/ijms25137375

**Published:** 2024-07-05

**Authors:** Junxiang Yan, Zixu Zhang, Hongdan Shi, Xinke Xue, Ang Li, Peng Ding, Xi Guo, Jinzhong Wang, Ying Wang, Boyang Cao

**Affiliations:** 1TEDA Institute of Biological Sciences and Biotechnology, Nankai University, Tianjin 300457, China; 1120210623@mail.nankai.edu.cn (J.Y.); 2120231637@mail.nankai.edu.cn (Z.Z.); 2120191115@mail.nankai.edu.cn (H.S.); 2120221548@mail.nankai.edu.cn (X.X.); dingpeng@nankai.edu.cn (P.D.); guoxi@nankai.edu.cn (X.G.); wangjinzhong@nankai.edu.cn (J.W.); 2Key Laboratory of Molecular Microbiology and Technology of the Ministry of Education, Nankai University, Tianjin 300457, China; 3Tianjin Key Laboratory of Microbial Functional Genomics, TEDA College, Nankai University, Tianjin 300457, China; 4State Key Laboratory of Medicinal Chemical Biology, College of Pharmacy, Tianjin Key Laboratory of Molecular Drug Research, Nankai University, Tianjin 300353, China; liang303@nankai.edu.cn

**Keywords:** *Plesiomonas shigelloides*, flagellar transcriptional hierarchy, RNA-seq, cross-talk, T6SS, T2SS-2, physiological and metabolic

## Abstract

*Plesiomonas shigelloides*, a Gram-negative bacillus, is the only member of the Enterobacteriaceae family able to produce polar and lateral flagella and cause gastrointestinal and extraintestinal illnesses in humans. The flagellar transcriptional hierarchy of *P. shigelloides* is currently unknown. In this study, we identified FlaK, FlaM, FliA, and FliA_L_ as the four regulators responsible for polar and lateral flagellar regulation in *P. shigelloides*. To determine the flagellar transcription hierarchy of *P. shigelloides*, the transcriptomes of the WT and Δ*flaK*, Δ*flaM*, Δ*fliA*, and Δ*fliA_L_* were carried out for comparison in this study. Quantitative Real-Time Polymerase Chain Reaction (qRT-PCR) and luminescence screening assays were used to validate the RNA-seq results, and the Electrophoretic Mobility Shift Assay (EMSA) results revealed that FlaK can directly bind to the promoters of *fliK*, *fliE*, *flhA*, and *cheY*, while the FlaM protein can bind directly to the promoters of *flgO*, *flgT*, and *flgA*. Meanwhile, we also observed type VI secretion system (T6SS) and type II secretion system 2 (T2SS-2) genes downregulated in the transcriptome profiles, and the killing assay revealed lower killing abilities for Δ*flaK*, Δ*flaM*, Δ*fliA*, and Δ*fliA_L_* compared to the WT, indicating that there was a cross-talk between the flagellar hierarchy system and bacterial secretion system. Invasion assays also showed that Δ*flaK*, Δ*flaM*, Δ*fliA*, and Δ*fliA_L_* were less effective in infecting Caco-2 cells than the WT. Additionally, we also found that the loss of flagellar regulators causes the differential expression of some of the physiological metabolic genes of *P. shigelloides*. Overall, this study aims to reveal the transcriptional hierarchy that controls flagellar gene expression in *P. shigelloides*, as well as the cross-talk between motility, virulence, and physiological and metabolic activity, laying the groundwork for future research into *P. shigelloides*’ coordinated survival in the natural environment and the mechanisms that infect the host.

## 1. Introduction

The genus *Plesiomonas*, represented by a single species, *Plesiomonas shigelloides*, is a Gram-negative bacillus associated with gastrointestinal and extraintestinal diseases in humans [1], such as an invasive shigellosis-like disease, cholera-like illness, pseudoappendicitis, meningitis, acute secretory gastroenteritis, and bacteremia [2,3,4,5,6,7]. Many pathogenic bacterial species express flagella on their surfaces, which are their primary means of locomotion or motility, and flagellar motility is necessary for them to reach the site of infection, which is the initial step in the establishment of a bacterial infection [8]. Meanwhile, flagellum motility provides a significant advantage for bacteria to move toward favorable conditions or away from detrimental environments and to effectively compete with other microorganisms [9].

The flagellum consists of three parts: the basal body, hook, and filament [10]. Flagellar motility, an energetically expensive process, requires the coordinated expression of more than 50 genes encoding regulatory proteins, structural components of the secretion and assembly apparatus, and chemo-sensor machinery [8]. The coordinated expression of these promoters clusters gene transcription at three or four levels of hierarchy: classes I to III or I to IV [11]. Peritrichous flagellated bacteria, such as *Salmonella* and *Escherichia coli*, have three levels of hierarchy, and transcription of class I and II genes requires the housekeeping sigma factor 70 (σ^70^) [12]. However, some bacteria, such as *Vibrio cholerae* [13,14,15], *Vibrio parahaemolyticus* [16], and *Aeromonas hydrophila* [11,17], have four levels of flagella regulatory hierarchy. Furthermore, several of the aforementioned bacteria contain both polar and lateral flagella. Bacteria with polar flagellation exhibit four transcriptional levels: class I promoters encode a σ^54^-dependent activator (FlrA) and activate class II σ^54^-dependent promoters (FlrBC), which encode a two-component signal-transducing system whose regulator activates class III σ^54^-dependent promoters [13,14,15,16,17]. Moreover, class II promoters also encode the σ^28^ factor (FliA), which activates the transcription of class IV genes. The lateral flagellar system, such as that of *V. parahaemolyticus* and *A. hydrophila*, which have three levels of flagella regulatory hierarchy, is encoded by more than 30 lateral flagella genes and regulated by the LafK master regulator [18]. Furthermore, LafK may compensate for FlaK’s role in enabling swimming in some bacteria [19]. In the expression of the lateral flagellar system, the master regulator LafK activates the class II genes, and then the expression of class III genes is activated by σ^28^, which is encoded by *fliA_L_* [16,20,21]. In a word, in a spatiotemporal-dependent manner, the expression of flagellar genes is strictly regulated by the respective regulators and sigma factors.

The primary function of flagella is to provide motility to bacteria; nevertheless, they are also related to bacterial pathogenicity. Flagellum-mediated motility plays a critical role in interactions between *Vibrio* species and their hosts, and these bacteria colonization and lethality in their respective hosts decrease when motility is restricted [22]. *Vibrios* have therefore been shown to be valuable models for researching flagellar control and its role in colonization. Motility has been identified as a virulence determinant of *V. cholerae*. Additionally, non-motile mutants in competition assays using an infant mouse model system show no significant defect in their capacity to colonize the small intestine [23]. A Na^+^ gradient across the membrane drives *V. cholerae* flagellar rotation while also regulating transcription of the *toxT* gene, which is necessary for CT and TCP expression, revealing a possible mechanism for connecting virulence factor expression to motility [24,25]. Coster and Kenner found that the human reactogenicity of live attenuated *V. cholerae* vaccines is also lowered in non-motile mutants [26]. In addition to being more frequently documented in *V. cholerae*, the correlation between flagella and virulence has also been investigated in other bacteria. Flagella motility, like that of *V. cholerae*, is critical in the pathogenesis of *V. campbellii*, and inhibiting motility greatly reduces host mortality during infection [27]. Previous studies have also reported the synergistic action of SPI-1 gene expression in *Salmonella enterica serovar typhimurium* through transcriptional cross-talk with the flagellar system [28]. Zhao et al. found that the type III secretion system intersects with the lateral flagellar system in *A. hydrophila* [29]. In a burned mouse model of infection, non-flagellated mutants of *P. aeruginosa* have been demonstrated to exhibit reduced penetration of cultured corneal epithelial cells and to be deficient in pathogenicity [30,31]. As a result, the significance of flagella in pathogenicity is obvious.

In addition to granting motility to bacteria and influencing adherence and virulence, flagellar regulators and flagellar genes are also involved in the physiological metabolism of bacteria. The relationship between the ability to survive periods of acid shock and the fine-tuned expression of motility genes and fitness is currently unknown; however, Schumacher et al. found an inverse correlation between the expression of FlhDC, the master regulator of flagellation, and acid shock survival in *E. coli* [32]. According to Hoque et al., the ∆*flrA* mutation in *V. cholerae* increases iron utilization and oxidative stress tolerance since it significantly upregulates most of the genes involved in heme absorption, siderophore uptake, and vibriobactin uptake [33]. Sinha-Ray et al. found that a mutation in *V. cholerae*’s *flrA* gene is inversely involved in the vps-independent biofilm driving bacteria toward nutrients in lake water [34]. Rodríguez-Herva et al. suggested that in addition to a number of motility and chemotaxis genes, the *fliA* gene product is also necessary for the expression of some genes potentially involved in amino acid utilization or stress responses [35]. Li et al. reported that flagellum hook-associated protein (FlgK) increases the resilience of *C. sakazakii* to dehydration and regulates formate dehydrogenase and nitrate reductase [36]. It has also been reported in studies that FliZ plays a key role in controlling lipase and hemolysin activities [37,38].

*P. shigelloides* is a special member of the Enterobacteriaceae family that can produce both polar and lateral flagella, and Merino et al. showed that *P. shigelloides* has two distinct gene clusters: one for lateral flagella biosynthesis and another for biosynthetic polar flagella [39]. Merino and co-workers reported that although *P. shigelloides* lacks a FlrB ortholog, *P. shigelloides* FlrC (FlaM) contains the PAS and His Kinase A domains found in the FlrC proteins of *Vibrio* species and *A. hydrophila*, indicating that *P. shigelloides* FlaM may activate transcription from σ^54^-dependent promoters of class III genes [39]. Furthermore, the *P. shigelloides* polar flagella gene regions show greater similarity to those reported in *Vibrio* or *Aeromonas* than the regions in Enterobacteriaceae [40]. Though a number of studies have implicated motility as being important for bacteria, the transcriptional regulation network of polar and lateral flagella in *P. shigelloides* has yet to be revealed. In this work, we identified FlaK, FlaM, FliA, and FliA_L_ as the four regulators responsible for polar and lateral flagellar regulation in *P. shigelloides*. Through transcriptome sequencing analysis, we confirmed the polar and lateral flagellar regulatory hierarchy and further revealed the cross-talk between the flagellar hierarchy system and *P. shigelloides*’ virulence and physiological metabolism.

## 2. Results

### 2.1. RNA Sequencing of Flagellar Regulatory Mutants

Given the basis of the previous research, we aligned the *P. shigelloides* flagellar gene clusters for orthologous alignment. The results indicated that the orthologous similarity of the flagellar gene in *P. shigelloides* with *V. cholerae*, *V. parahaemolyticus*, and *A. hydrophila* is relatively similar (Appendix A). Moreover, the highest overall homologous similarity is between the polar and lateral flagella gene clusters of *P. shigelloides* and *V. parahaemolyticus*, followed by *A. hydrophila* and *V. cholerae*. Previous reports corroborate our observations: The polar flagella gene cluster in the *P. shigelloides* genome contains the regulatory factors FlaK, FlaM, and FliA. Similar to FlrBC in *Vibrio* sp. or *A. hydrophila*, FlaM serves a similar function. Furthermore, the major regulator LafK is absent from the *P. shigelloides* lateral flagella gene cluster, which only contains the gene FliA_L_. To determine which *P. shigelloides* flagella genes are controlled by the flagellar transcription hierarchy, RNA-Seq of the WT and *flaK*, *flaM*, *fliA*, and *fliA*_L_ deletion strains was carried out for comparison. The volcano plot revealed that FlaK, FlaM, FliA, and FliA_L_ regulate the expression of 152, 341, 515, and 37 differentially expressed genes (DEGs) in their respective transcriptomes (Figure 1A–D). Moreover, the DEGs’ functionalities and enriched pathways in the Δ*flaK*, Δ*flaM*, Δ*fliA*, and Δ*fliA_L_* mutants were analyzed using the Gene Ontology (GO) and Kyoto Encyclopedia of Genes and Genomes (KEGG) pathway databases (Appendix A). Additionally, three upregulated and three downregulated DEGs in the Δ*flaK*, Δ*flaM*, Δ*fliA*, and Δ*fliA_L_* transcriptome profiles were selected, respectively, for validation using qRT-PCR in the WT and mutant strains. The results of qRT-PCR were consistent with RNA-seq analysis (Figure 1E–H), indicating the reliability of the RNA-seq.

### 2.2. Refinement of the Flagellar Transcription Hierarchy

*P. shigelloides* contains over 50 polar flagellar genes and up to 36 lateral flagellar genes (Appendix A). The heat map of polar flagellar gene expression in the Δ*flaK*, Δ*flaM*, Δ*fliA*, and Δ*fliA_L_* transcriptome profiles revealed that FlaK positively regulated nearly all of the polar flagellar gene clusters (Figure 2A and Appendix A), which also indicates that FlaK, as a primary regulator of the polar flagellar hierarchy, is important for the regulation of *P. shigelloides*’ motility. On various promoters of the polar flagellar gene cluster, FlaM and FliA, however, have distinct regulatory functions. The transcriptome data indicate that FlaM positively regulates the expression of polar flagellar gene clusters *flgAMN*, *flgB-J*, *flgKL*, and *flgOP* and flagellar gene *flgT*; however, the expression of *fliE-R*, a polar flagellar gene cluster, was upregulated following the loss of FlaM (Figure 2A and Appendix A). Furthermore, the results also revealed that FliA positively regulated the expression of the flagellar gene *flaC*, flagellar gene cluster *flaGHIIJ*, and chemotaxis genes *cheVR*, *motAB*, and *pomAB*, while FliA had a negative regulatory effect on the flagellar gene cluster that FlaM positively regulated (Figure 2A and Appendix A). To confirm the authenticity of the RNA-seq results, the flagellar genes positively regulated by FlaK, FlaM, and FliA were verified, respectively, by using qRT-PCR and construct promoter–lux fusions; the results of qRT-PCR and lux assays were consistent with RNA-seq (Figure 2C–E). Furthermore, consistent with previous reports, we also found that *P. shigelloides’* genome lacked the master regulator LafK in the lateral flagellar system; yet, LafK in *V. parahaemolyticus* and *A. hydrophila* exhibited high homologous similarity to FlaK in *P. shigelloides* (Appendix A). Moreover, we found that the downregulation of lateral flagella genes coincided with the loss of Flak (Appendix A), suggesting that FlaK may be involved in the regulation of the lateral flagellar system. For this reason, we performed a heatmap analysis of the lateral flagella gene expression in the transcriptome profiles of Δ*flaK* and Δ*fliA_L_* (Figure 2B). The results revealed that FliA_L_ positively regulated the expression of the lateral flagellar gene *fliC*_L_ and the lateral flagellar gene clusters *flgMN*_L_, *flgKL*_L_, and *fliDS*_L_-*lafX*-*fliKL*_L_-*fliA*-*lafTU* (Figure 2B and Appendix A). The differential expression of the lateral flagellar gene in the transcriptome profiles of Δ*flaK* and Δ*fliA_L_* was then validated by qRT-PCR and construct promoter–lux fusions (Figure 2F,G), which demonstrated that the experimental data matched.

Furthermore, the EMSAs results showed that FlaK can directly bind to the promoters of *fliK*, *fliE*, *flhA*, and *cheY*, but not to the promoters of *flhB* and *flaM* (Figure 2H). The FlaM protein can bind directly to the promoters of *flgO*, *flgT*, and *flgA* but not those of *flgB*, *flgK*, and *flgL* (Figure 2I). Unfortunately, no flagellar gene promoter was screened for binding to FliA or FliA_L_. In addition, we observed the migration of the WT, Δ*flaK*, Δ*flaM*, Δ*fliA*, Δ*fliA_L_*, and complementation strains grown in swimming agar plates and the flagella produced by the aforementioned strains, which indicated that Δ*flaK*, Δ*fliA*, and Δ*fliA_L_* have reduced motility compared with the WT, while Δ*flaM* completely loses motility, and the complementation strains recovered the motility level of the WT (Figure 2J). The TEM results showed that the creation of flagella decreased and grew shorter in Δ*flaK*, Δ*flaM*, and Δ*fliA*, while Δ*flaM* was generated without any flagella, and the flagella production ability of corresponding complementation strains was greatly increased (Figure 2K). Together with our earlier research on the regulation of *P. shigelloides* flagellar genes by RpoN, we also demonstrated that FlaK and FlaM in *P. shigelloides* are σ^54^-dependent activators, which is similar to *V. parahaemolyticus* and *A. hydrophila*. Finally, we characterized and mapped the polar and lateral flagellar gene transcriptional hierarchy of *P. shigelloides* (Figure 2L,M), drawing from a series of observed experimental results. Taken together, the revelation of the lateral and polar flagella transcriptional hierarchy of *P. shigelloides*, a pathogenic bacterium, not only completes the flagella gene transcriptional regulation network but also paves the way for its processes of motility and chemotaxis.

### 2.3. Flagellar Regulatory Mutants Reduce the Killing Ability and Virulence of P. shigelloides to Varying Degrees

In this study, the differentially expressed genes are also involved in the bacterial secretion system, including the observed T6SS significantly downregulated in the Δ*flaK*, Δ*flaM*, and Δ*fliA_L_* transcriptome profiles (Figure 3A) and T2SS-2 downregulated in the Δ*flaM* and Δ*fliA* transcriptome profiles (Figure 3B). Meanwhile, we validated the differential expression of T6SS or T2SS-2 genes in each transcriptome using qRT-PCR in the WT, Δ*flaK*, Δ*flaM*, Δ*fliA*, Δ*fliA_L_*, and complementation strains (Figure 3C–G). The qRT-PCR results were consistent with RNA-seq, which also suggests a cross-regulatory relationship between the flagellar system and bacterial secretion system. T6SS is a well-known contact-dependent bacterial weapon that can kill competitors directly by translocating proteinaceous toxins. Furthermore, we confirmed in a prior publication that T2SS-2 is also associated with *P. shigelloides*’ killing ability. Subsequently, the killing assay was used to compare the ability of the WT, Δ*flaK*, Δ*flaM*, Δ*fliA*, Δ*fliA_L_*, and complementation strains to kill *E. coli* MG1655. The results showed that the killing abilities of Δ*flaK*, Δ*flaM*, Δ*fliA*, and Δ*fliA_L_* were all decreased compared to the WT, while the killing abilities of the relative complement strains were restored to the level of the WT (Figure 3H). However, the killing ability of Δ*flaM* was significantly decreased, which may be because the absence of FlaM affected the expression of two gene clusters, T6SS and T2SS-2. Furthermore, invasion assays were performed to verify whether FlaK, FlaM, FliA, and FliA_L_ affect the virulence of *P. shigelloides*. Subsequently, invasion experiments revealed that, in comparison with the WT, Δ*flaK*, Δ*flaM*, Δ*fliA*, and Δ*fliA_L_* were all less able to infect Caco-2 cells, whereas Δ*flaM* was significantly reduced, and the relative complement strains’ ability to infect Caco-2 cells was restored to WT levels (Figure 3I). However, we did not find any transcript-level changes in the T3SS gene in the transcriptome. We speculated that FlaK, FlaM, FliA, and FliA_L_ may also affect the ability of *P. shigelloides* to infect Caco-2 cells by upregulating the expression of T6SS or T2SS-2, or their absence may cause *P. shigelloides’* motility to decrease and thus affect the infectability of *P. shigelloides*. Taken together, our findings unveil a connection between the flagellar and secretory systems in *P. shigelloides* and provide new insights into the pathogenesis of this bacterium.

### 2.4. Cross-Talk between the Flagellar System and P. shigelloides’ Physiological Metabolism

In this study, we also found that the loss of flagellar regulators causes the differential expression of some of the physiological metabolic genes of *P. shigelloides*, and our proposal for the cross-talk between the flagellar system and *P. shigelloides*’ physiological metabolism is presented in Figure 4.

## 3. Discussion

*Plesiomonas shigelloides*, a Gram-negative foodborne pathogen of the Enterobacteriaceae family, has been known to exist for almost 80 years since Ferguson’s 1947 discovery [41]. *P. shigelloides* is extensively present in freshwater lakes, rivers, and streams and can be found in the environment, in animals, and in humans [42,43]. Acute gastroenteritis is the primary sign of infection; it can also lead to sepsis and neurological conditions [42]. *P. shigelloides* has received comparatively less attention and reports than other foodborne pathogens such as *Salmonella*, *V. parahemolyticus*, and diarrheal *E. coli* [44]. For the majority of pathogenic bacteria, the flagellum is a necessary component of their motility as well as their pathogenicity, which allows them to adhere to, colonize, or infect eukaryotic cells.

Currently, the flagellar hierarchical regulatory systems of most pathogens are reported, either *E. coli* and *Salmonella*, which have tertiary flagellar hierarchical regulation, or *V. cholerae* [13,14,15], *V. parahaemolyticus* [16], *A. hydrophila* [11,17], and *P. aeruginosa* [8] with fourth flagellar regulation. Two different *P. shigelloides* flagella gene clusters were described previously [39], and combined with the previously reported [40] and orthologous similarity alignment to the flagellar gene clusters, we found that the polar and lateral flagella gene clusters of *P. shigelloides* and *V. parahaemolyticus* exhibit the greatest overall homologous similarity, trailed by *A. hydrophila* and *V. cholerae*. In this study, RNA-Seq, qRT-PCR, lux assay, and EMSAs were used to confirm the flagellar transcription hierarchy. The EMSA results revealed that FlaK can directly bind to the promoters of *fliK*, *fliE*, *flhA*, and *cheY*, while the FlaM protein can bind directly to the promoters of *flgO*, *flgT*, and *flgA*, but no flagellar gene promoter was tested for binding to FliA or FliA_L_. The polar flagellate hierarchy system regulated by FlaK, FlaM, and FliA was finally confirmed. And we also demonstrated that FlaK and FlaM in *P. shigelloides* are σ^54^-dependent activators in earlier research on the regulation of *P. shigelloides*’ flagellar genes by RpoN [45]. In the *P. shigelloides* lateral gene cluster, only FliA_L_ was present, but no LafK ortholog was found. All lateral gene clusters, including the non-functional ones in the Enterobacteriaceae, have reported this gene [46]. On the other hand, LafK in *V. parahaemolyticus* and *A. hydrophila* showed high homologous similarity to FlaK in *P. shigelloides*, and the deletion of *flaK* results in the downregulation of lateral flagella gene clusters in RNA-seq, qRT-PCR, and lux assays, indicating that it is possible that FlaK regulates the lateral flagellar system. On this premise, we defined the lateral flagellar gene transcriptional hierarchy in *P. shigelloides* under the regulation of FlaK and FliA_L_; nevertheless, FlaK may replace LafK’s function in regulating the lateral flagellar system, which requires additional validation. Furthermore, we also observed that the upregulation of flagellar genes in the RNA-seq data, which we hypothesized was caused by the complex flagellar regulation mechanism of *P. shigelloides*, and it is possible that there is a negative regulation of flagellar genes, for which we have further investigation. Additionally, we will further confirm the binding sites of FlaK and FlaM in the promoter region of the flagellar gene using DNA footprinting.

It was previously reported that there was cross-talk between the flagellar hierarchy system and the bacterial secretion system. Speare L et al. demonstrated that flagella-dependent aggregation factors and TasL (T6SS2) may act in concert to facilitate cell–cell contact, thus mediating the killing ability of bacteria [47]. In the early stages of infection, Huang Z et al. demonstrated that the lateral flagellar-associated *flhA* gene was crucial for the adhesion and colonization of *V. metschnikovii*. Additionally, they also discovered that *V. metschnikovii* promoted a high level of cytotoxicity through the synergistic interaction of T6SS and the lateral flagella [48]. Furthermore, Bouteiller M et al. demonstrated that there was a cross-talk between *Pseudomonas fluorescens*’s class IV flagellar gene expression and the type VI secretion system [49]. In this study, we observed T6SS significantly downregulated in the Δ*flaK*, Δ*flaM*, and Δ*fliA_L_* transcriptome profiles and T2SS-2 downregulated in the Δ*flaM* and Δ*fliA* transcriptome profiles, which were then validated by qRT-PCR. Previous work has indicated a correlation between T2SS-2 and the killing ability of *P. shigelloides* [45]. In the meantime, the killing assay revealed lower killing abilities for Δ*flaK*, Δ*flaM*, Δ*fliA*, and Δ*fliA_L_* compared to the WT, indicating a cross-regulatory relationship between the flagellar and bacterial secretion systems in *P. shigelloides*. Furthermore, a number of investigations also showed a connection between the flagellar system and T3SS, which influences the pathogenicity of bacteria. In *P. plecoglossicida*, Qin et al. found cross-talk between the type III secretion system and flagella assembly using comparative secretome analysis [50]. Soria-Bustos J et al. suggested that the presence of an intact T3SS is required for the assembly of flagella, highlighting the existence in the EPEC of a cross-talk between these two virulence-associated T3SSs [51]. Nevertheless, while our invasion experiments revealed that Δ*flaK*, Δ*flaM*, Δ*fliA*, and Δ*fliA_L_* were less capable of infecting Caco-2 cells than the WT, we were unable to detect any transcript-level alterations of the T3SS gene in the transcriptome. And the growth assays in the lag and log phases also showed no significant difference between the WT and Δ*flaK*, Δ*flaM*, Δ*fliA*, and Δ*fliA_L_* strains (Appendix A). In this case, we hypothesized that the downregulation of T6SS and T2SS-2, or the expression of some virulence genes caused by the loss of flagella regulators in this study, or the decreased motility of *P. shigelloides* affected the adhesion of *P. shigelloides*, resulting in a decrease in *P. shigelloides* infection to Caco-2 cells, which, of course, requires further confirmation.

In addition to the flagellar system and the bacterial secretion system, we have also previously described the correlation between the flagellar system and bacterial physiological metabolism [32,33,34,35,36,37,38]. Similarly, in this study, we also found that the loss of flagellar regulators causes the differential expression of some of the physiological metabolic genes of *P. shigelloides*. Among the genes downregulated in the Δ*flaK* mutant were genes mainly responsible for periplasmic nitrate reductase, pyrimidine metabolism, and purine metabolism (Appendix A). Upregulated genes in the Δ*flaK* mutant included those involved in myo-inositol catabolism and propanoate metabolism (Appendix A). FlaM regulates slightly more physiological and metabolic genes than FlaK does. Among the genes downregulated in the Δ*flaM* mutant were those responsible for the biosynthesis of amino acids, glycerophospholipid metabolism, the citrate cycle, the ribosome, purine metabolism, and thiamine metabolism (Appendix A). Among the genes upregulated in the Δ*flaM* mutant were those responsible for hydrogenase (Appendix A). The physiological metabolic genes regulated by FliA are the most numerous of the four flagellar hierarchy regulators. Among the genes downregulated in the Δ*fliA* mutant were those responsible for the citrate cycle, fatty acid degradation and metabolism, glyoxylate and dicarboxylate metabolism, periplasmic nitrate reductase, NADH-quinone oxidoreductase, pyrimidine metabolism, hydrogenase, glycerophospholipid metabolism, and curli production assembly (Appendix A). Among the genes upregulated in the Δ*fliA* mutant were those responsible for glycolysis/gluconeogenesis, ribosome, myo-inositol catabolism, pyruvate metabolism, starch and sucrose metabolism, and porphyrin metabolism (Appendix A). Both FliA in the polar flagellar gene cluster and FliA_L_ in the lateral flagellar gene cluster are σ factors; however, the number of DEGs regulated by both differs greatly. Among the genes downregulated in the Δ*fliA_L_* mutant were genes responsible for periplasmic nitrate reductase (Appendix A). Among the genes upregulated in the Δ*fliA_L_* mutant were those responsible for myo-inositol catabolism (Appendix A). Although we revealed a relationship between the flagellar system and *P. shigelloides*’ physiological metabolic activity, the mechanism requires more investigation (Figure 4).

Overall, our study not only revealed the transcriptional hierarchy of polar and lateral flagellar genes but also demonstrated the cross-talk between the flagellar system and the secretory system in *P. shigelloides*, which influences *P. shigelloides*’ killing ability. Additionally, we also identified the effects of the flagellar system on the virulence and partial physiological metabolism of *P. shigelloides*. These findings will provide a foundation for understanding the relationship and mechanism between motility, virulence, and physiological and metabolic activity in *P. shigelloides*.

## 4. Materials and Methods

### 4.1. Bacterial Strains, Plasmids, and Growth Conditions

Appendix A shows the plasmids and bacterial strains used in this study. According to the experimental requirements, bacteria were cultured at 30 °C or 37 °C using Luria–Bertani (LB) liquid, semi-solid media, and solid, as well as Dulbecco’s Modified Eagle’s Media (DMEM) supplemented with 20% fetal bovine serum (FBS). Meanwhile, supplements containing 25 μg/mL of ampicillin, 50 μg/mL of kanamycin, or 25 μg/mL of chloramphenicol were added to the media as needed.

### 4.2. Construction of Gene Deletion Strains and Complementation

The gene deletion methods in this study were performed as previously reported [52]. To put it briefly, the target genes’ (*flaK*, *flaM*, *fliA*, and *fliA_L_*) upstream and downstream regions were amplified using the genome of the *P. shigelloides* WT as the PCR amplification template. These upstream and downstream regions were then connected by PCR amplification, and the successfully connected DNA fragments and the pRE112 plasmid were digested by an endonuclease and linked with DNA ligase to form a new recombinant vector. The recombinant vector was electrotransformed into *E. coli* S17-1 λpir and subsequently mixed with the *P. shigelloides* WT equally in antibiotic-free solid plates at 37 °C for 24 h for homologous recombination. Mixed-grown colonies were subsequently diluted and plated onto a medium containing 25 μg/mL ampicyl and chloramphenicol, and agarose gel electrophoresis and PCR were used to identify the positive single colonies. Ultimately, the positively identified clones were transferred to solid plates with 20% sucrose and a 25 μg/mL ampicyl antibiotic concentration. The pRE112 plasmid’s susceptibility to sucrose-induced lethality and the inability of the *P. shigelloides* WT to grow in a medium containing chloramphenicol were used to screen for gene deletion mutant strains. The complementation strains, pBAD33/*flaK*^+^, pBAD33/*flaM*^+^, pBAD33/*fliA*^+^, and pBAD33/*fliA_L_*^+^, were constructed by introducing the recombinant vector, pBAD33 plasmid with target genes, into the relative gene-deleted strains via electroporation. Agarose gel electrophoresis and PCR product sequencing were used to verify the accuracy of the deletion and complementation strains. All primers used in this study are listed in Appendix A.

### 4.3. RNA Isolation and Transcriptome Sequencing

The WT and deletion mutant strains (Δ*flaK*, Δ*flaM*, Δ*fliA*, and Δ*fliA_L_*) were cultured overnight at 37 °C after diluting 1:100 into 20 mL of LB liquid medium, and then the bacteria were transferred to a fresh medium the next day and grew to OD600 = 0.8 (each strain was transduced to three culture media simultaneously). TRIzol^®^ Reagent (Invitrogen, Waltham, MA, USA) was used to separate total RNA from the WT and deletion mutant strains (three total RNA samples were extracted from each strain), which was subsequently processed with RNase-free DNase and dissolved in RNase-free water. Meanwhile, RNA degradation and contamination were assessed using 1% agarose gel, and a NanoDrop-2000 spectrophotometer (Thermo Fisher, Waltham, MA, USA) measured RNA concentration at OD260 and determined RNA purity via the ratio of OD260/OD230 and OD260/OD280. Following testing and qualifying, the cDNA library was sequenced on an Illumina NovaSeq 6000 platform (Illumina, San Diego, CA, USA) to produce 150 bp paired-end reads. Gene expression levels were measured using HTSeq, and the length and number of reads mapped to each gene were used to calculate the Fragments Per Kilobase of transcript sequence per Million base pairs sequenced (FPKM) value of each gene. The criteria for differentially expressed genes (DEGs) were set as |log2 fold change| ≥ 1 and adjusted *p*-value (padj) ≤ 0.05.

### 4.4. Quantitative Real-Time Polymerase Chain Reaction (qRT-PCR)

To further validate the RNA-Seq data and reveal the flagellar hierarchy regulatory network, we performed qRT-PCR for the WT, Δ*flaK*, Δ*flaM*, Δ*fliA*, Δ*fliA_L_*, pBAD33/*flaK*^+^, pBAD33/*flaM*^+^, pBAD33/*fliA*^+^, and pBAD33/*fliA_L_*^+^. The isolation, quantification, and detection of total RNA from the WT, deletion mutant strains, and complementation strains were performed as described above. cDNA was generated using the PrimeScript™ RT reagent kit (Takara Bio, Shiga, Japan), and qRT-PCR analysis was performed using an Applied Biosystems ABI 7500 sequence detection system (Applied Biosystems, Foster City, CA, USA). A 96-well optical reaction plate (Applied Biosystems) was used for each qRT-PCR experiment, which included 1 μL of cDNA, 10 μL of FastStart Universal SYBR Green Master (ROX) mix, and two gene-specific primers with a final concentration of 0.3 mM each. The cycle threshold approach (2^−∆∆CT^) was used to compute relative target gene expression levels as fold changes [53], with the *gyrB* gene in *P. shigelloides* serving as a reference control [45]. Furthermore, several steps must be taken to reduce qRT-PCR error during the experiment, such as preventing RNA degradation and DNA contamination, accurately quantifying RNA, using deionized water as a negative control template to avoid non-specific amplification, and selecting appropriate internal reference genes. Each experiment was conducted three times.

### 4.5. Motility Assay and Transmission Electron Microscopy (TEM) of Flagella

The motility assay was carried out as previously described [54]. In short, the WT, Δ*flaK*, Δ*flaM*, Δ*fliA*, Δ*fliA_L_*, pBAD33/*flaK*^+^, pBAD33/*flaM*^+^, pBAD33/*fliA*^+^, and pBAD33/*fliA_L_*^+^ strains were inoculated into 20 mL of LB liquid medium and cultured overnight at 37 °C, and then transferred to fresh medium the next day and cultured to OD600 = 0.8. Subsequently, 1 μL of the fresh bacterial solution was absorbed into the center of the swimming agar plates and static cultured for 12 h at 30 °C, and then the bacteria’s swimming distance was measured and photographed. The trials were conducted at three different time points, each with six repetitions. Moreover, TEM and negative staining were used to visualize the flagella of the WT, Δ*flaK*, Δ*flaM*, Δ*fliA*, Δ*fliA_L_*, pBAD33/*flaK*^+^, pBAD33/*flaM*^+^, pBAD33/*fliA*^+^, and pBAD33/*fliA_L_*^+^ strains, as also previously described [55]. Briefly, all strains were inoculated into 20 mL of LB liquid medium and cultured overnight at 37 °C, and then transferred to fresh medium the next day and cultured to OD600 = 0.4. A 20 μL aliquot of each sample was separately placed on a Formvar/carbon grid (400 mesh), which was glow-discharged prior to use to increase the hydrophilicity. The grids were washed with 0.1 M sodium acetate (pH 6.6), negatively stained with 2% phosphotungstic acid, and air dried for 6 min before being viewed at 120 kV with a Tecnai transmission electron microscope (Thermo Fisher, Waltham, MA, USA).

### 4.6. Luminescence Screening Assay

Luminescence screening assays were carried out as previously described [56]. In brief, PCR was used to amplify the target gene’s promoter region, which was then digested by two restriction enzymes, XhoI and BamHI, along with plasmid pMS402, to generate a new recombinant plasmid using DNA ligase. Subsequently, the fusion reporter plasmid used in this study (Appendix A) was transformed into the WT and deletion mutant strains, and then the positive clones, screened and identified, were grown in the LB medium to the mid-log phase. The promoter activity was evaluated at OD600 with a Synergy 2 plate reader (Agilent BioTek, Santa Clara, CA, USA). Each experiment was conducted three times, with six replicates each.

### 4.7. Expression and Purification of Proteins and Electrophoretic Mobility Shift Assays (EMSAs)

The proteins used in this study’s EMSAs, FlaK, FlaM, FliA, and FliA_L_, were cloned into pMAL-c5X and expressed in *E. coli* BL21 (DE3) and purified using amylose resin (New England Bio Labs, Ipswich, MA, USA) affinity chromatography. EMSAs were performed using a mixture of each probe, 60 ng, and escalating doses of the fusion protein in a 20 μL reaction volume with 20 mM Tris-HCl (pH 7.5), 120 mM KCl, 5 mM MgCl_2_, 1 mM DTT, 5% glycerol, and 1 μg Poly (dI.dC) incubated at 30 °C for 45 min. Moreover, the DNA–protein complexes were separated using a 6% PAGE in a 0.5 × TBE buffer for 2 h at 140 V. After being stained with GelRed for 5 min, gels were scanned using a gel imaging system (GE Healthcare, Chicago, IL, USA).

### 4.8. Killing Assay

The *E. coli* MG1655 killing assay, with some modifications, was carried out as described previously [57]. Overnight cultures from *P. shigelloides* and *E. coli* were diluted in the LB medium and grown at 37 °C until the bacteria reached OD600 = 1.5, and then the cells were harvested and concentrated. The ratio of a 1:1 mixture of the predator and prey bacteria was prepared, and 20 µL of this mixture was added to antibiotic-free LB agar plates. The two bacteria were then co-cultured for 3 h, washed with phosphate buffer, and serial dilutions were spotted onto antibiotic-containing LB solid plate medium and incubated overnight at 37 °C. The killing ability of the Δ*flaK*, Δ*flaM*, Δ*fliA*, Δ*fliA_L_*, pBAD33/*flaK*^+^, pBAD33/*flaM*^+^, pBAD33/*fliA*^+^, and pBAD33/*fliA_L_*^+^ strains was reported as a percentage relative to that of the WT. The killing assay was performed three times, with six repetitions each time.

### 4.9. Growth Assay

Growth assays were performed as described previously [58]. In brief, the WT, Δ*flaK*, Δ*flaM*, Δ*fliA*, and Δ*fliA_L_* strains were cultured at 37 °C in an LB medium overnight, and the cultured bacterial solution, at a ratio of 1:200 per well, was added to a 96-well cell plate containing 200 μL of LB or DMEM. Meanwhile, the sterile LB and DMEM served as a control group. A Molecular Devices Spectramax 190 full-wavelength microplate reader (Molecular Devices LLC., San Jose, CA, USA) was used for the dynamic growth experiment. The trials were conducted at three different time points, each with six repetitions.

### 4.10. Invasion Assay

The invasion assay was performed with minor modifications to the previously described method [59]. Briefly, the WT, Δ*flaK*, Δ*flaM*, Δ*fliA*, Δ*fliA_L_*, pBAD33/*flaK*^+^, pBAD33/*flaM*^+^, pBAD33/*fliA*^+^, and pBAD33/*fliA_L_*^+^ strains were cultured overnight in LB, and then transferred to new LB the next day, at a 1:100 inoculation ratio, until the bacteria reached OD600 = 0.6. Confluent monolayers of roughly 1 × 10^5^ Caco-2 cells per well, in 24-well plates, were overlaid with approximately 5 × 10^7^ WT, Δ*flaK*, Δ*flaM*, Δ*fliA*, Δ*fliA_L_*, pBAD33/*flaK*^+^, pBAD33/*flaM*^+^, pBAD33/*fliA*^+^, and pBAD33/*fliA_L_*^+^ bacterial cells. Subsequently, Caco-2 and bacterial cells, at 37 °C in 5% CO_2,_ were co-cultured for 1 h to induce invasion, and then 100 μg/mL of gentamicin was added to the cell culture medium to kill extracellular bacteria. Finally, the cells, after the monolayer had been twice washed with PBS, were lysed for 10 min using 0.1% Triton X-100. The invasion rate was calculated as the ratio of the number of recovered bacteria to the total number of bacterial cells used for infection. Furthermore, the invasion assay was performed three times, with six repetitions each time.

### 4.11. Statistical Analysis

GraphPad Prism v7.0 software (GraphPad Inc., La Jolla, CA, USA) was used to statistically analyze all data, which are expressed as means ± standard deviation (SD) [60]. The independent samples *t*-test and Mann–Whitney *U* test were used to determine differences between the groups. Furthermore, a probability value (*p*) ≤ 0.05 was considered statistically significant (*** *p* ≤ 0.001; ** *p* ≤ 0.01; * *p* ≤ 0.05; ns indicates not significant).

## Figures and Tables

**Figure 1 ijms-25-07375-f001:**
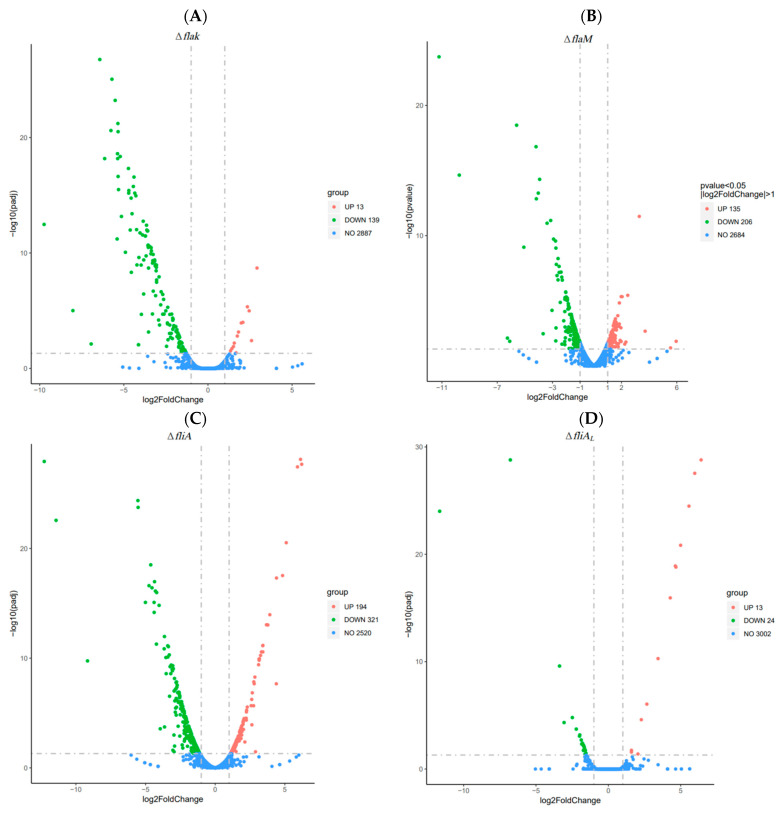
Transcriptomic analysis of the *P. shigelloides* between the WT and Δ*flaK*, Δ*flaM*, Δ*fliA*, and Δ*fliA_L_* strains. The volcano plot of differentially expressed genes (DEGs) in the Δ*flaK* (**A**), Δ*flaM* (**B**), Δ*fliA* (**C**), and Δ*fliA_L_* (**D**) transcriptome profiles. The red circle was upregulated genes, the green circle was downregulated genes, and the blue circle had no DEGs. Three upregulated and three downregulated DEGs were selected for validation using qRT-PCR in the Δ*flaK* (**E**), Δ*flaM* (**F**), Δ*fliA* (**G**), and Δ*fliA_L_* (**H**) transcriptome profiles, respectively.

**Figure 2 ijms-25-07375-f002:**
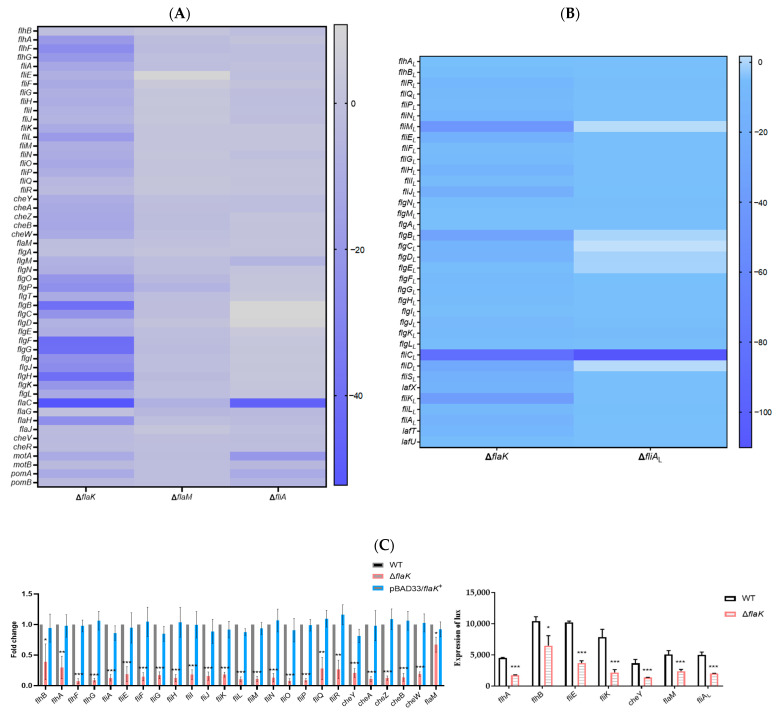
Flak, FlaM, FliA, and FliA_L_ regulated the motility and flagellar synthesis of *P. shigelloides.* (**A**) A heat map of polar flagellar gene expression in the Δ*flaK*, Δ*flaM*, and Δ*fliA* transcriptome profiles. (**B**) A heat map of lateral flagellar gene expression in the Δ*flaK* and Δ*fliA_L_* transcriptome profiles. (**C**) Transcription verification of the polar flagellar gene clusters II in the Δ*flaK* transcriptome profiles by qRT-PCR and lux assay. (**D**) Transcription verification of the polar flagellar gene clusters III in the Δ*flaM* transcriptome profiles by qRT-PCR and lux assay. (**E**) Transcription verification of the polar flagellar gene clusters IV in the Δ*fliA* transcriptome profiles by qRT-PCR and lux assay. (**F**) Transcription verification of the lateral flagellar gene clusters II in the Δ*flaK* transcriptome profiles by qRT-PCR and lux assay. (**G**) Transcription verification of the lateral flagellar gene clusters III in the Δ*fliA_L_* transcriptome profiles by qRT-PCR and lux assay. (**H**) The EMSAs of the Flak protein with *fliK* and *fliE*, *flhA* and *cheY*, and *flhB* and *flaM* promoters. (**I**) The EMSAs of the FlaM protein with *flgO* and *flgT*, *flgA* and *flgB*, and *flgK* and *flgL* promoters. (**J**) The motility of the WT, Δ*flaK*, Δ*flaM*, Δ*fliA*, Δ*fliA_L_*, and complementation strains grown in swimming agar plate. (**K**) A TEM visualization of the flagella produced by the WT, Δ*flaK*, pBAD33/*flaK^+^,* Δ*flaM*, pBAD33/*flaM*^+^, Δ*fliA*, pBAD33/*fliA^+^,* Δ*fliA_L_*, and pBAD33/*fliA_L_*^+^. The hollow bacterial flagella were pointed to by the colored arrows. (**L**) The polar flagellar gene transcriptional hierarchy of *P. shigelloides*. (**M**) The putative lateral flagellar gene transcriptional hierarchy of *P. shigelloides*. Significant differences were indicated by asterisks (*** *p* ≤ 0.001; ** *p* ≤ 0.01; * *p* ≤ 0.05).

**Figure 3 ijms-25-07375-f003:**
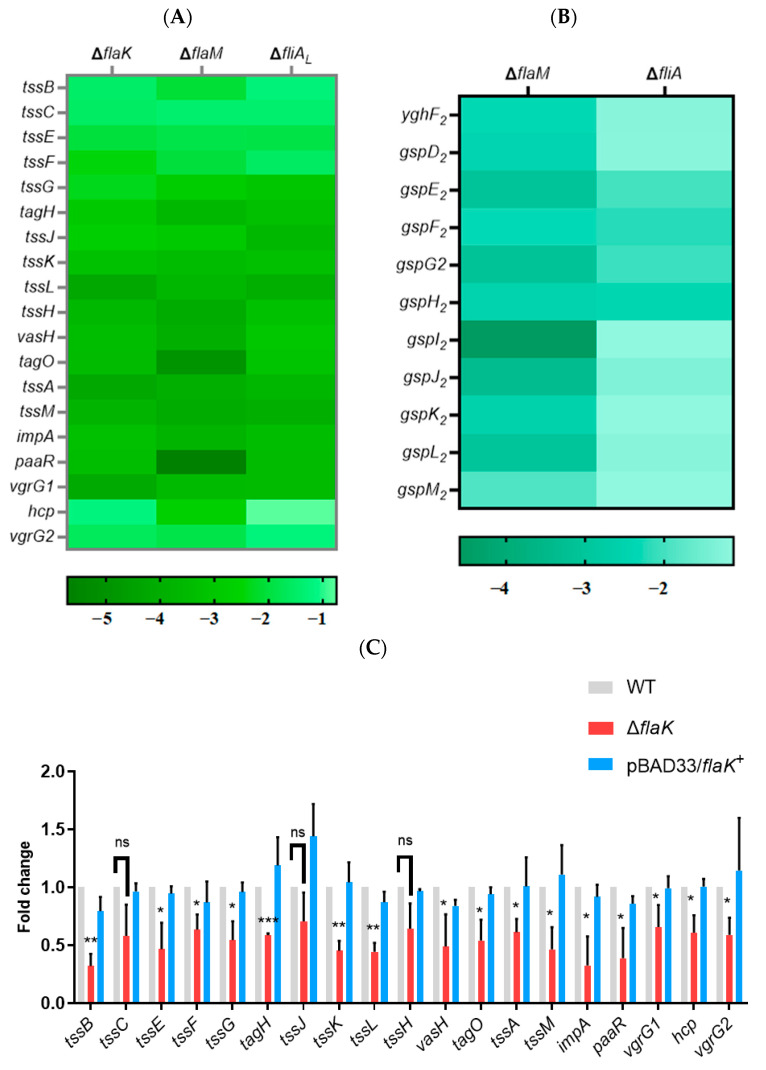
(**A**) A heat map of the T6SS gene cluster expression in the Δ*flaK*, Δ*flaM*, and Δ*fliA_L_* transcriptome profiles. (**B**) A heat map of the T2SS-2 gene cluster expression in the Δ*flaM* and Δ*fliA* transcriptome profiles. (**C**) Transcription verification of the T6SS gene clusters in the Δ*flaK* transcriptome profiles by qRT-PCR. (**D**) Transcription verification of the T6SS gene clusters in the Δ*flaM* transcriptome profiles by qRT-PCR. (**E**) Transcription verification of the T2SS-2 gene clusters in the Δ*flaM* transcriptome profiles by qRT-PCR. (**F**) Transcription verification of the T6SS gene clusters in the Δ*fliA_L_* transcriptome profiles by qRT-PCR. (**G**) Transcription verification of the T2SS-2 gene clusters in the Δ*fliA* transcriptome profiles by qRT-PCR. (**H**) The killing assay of the WT, Δ*flaK*, Δ*flaM*, Δ*fliA*, Δ*fliA_L_*, and the complementation strains. (**I**) The invasion assay of the WT, Δ*flaK*, Δ*flaM*, Δ*fliA*, Δ*fliA_L_*, and the complementation strains. Significant differences were indicated by asterisks (*** *p* ≤ 0.001; ** *p* ≤ 0.01; * *p* ≤ 0.05; ns indicates not significant).

**Figure 4 ijms-25-07375-f004:**
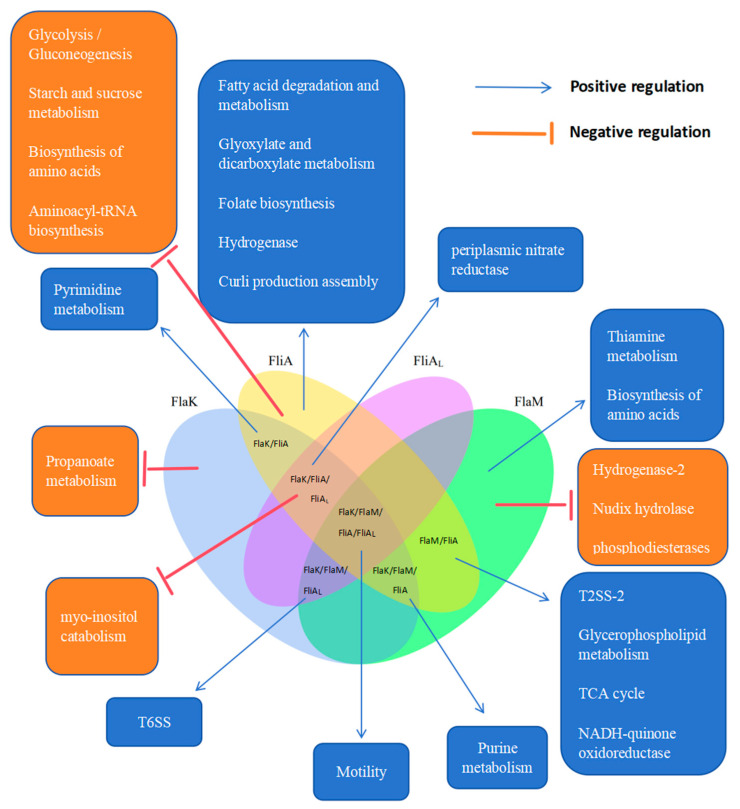
A schematic of the proposed cross-talk between the flagellar system and *P. shigelloides*’ physiological metabolism. The light blue, yellow, purple, and light green regions in the venn diagram represent transcriptomic profiles affected by Flak, FliA, FliA_L_, and FlaM, respectively.

## Data Availability

The RNA sequencing data generated in this study are available in the NCBI SRA database (accession numbers: SRR22678961, SRR22679947, SRR22696781, SRR22686293, and SRR22686286). Other data are presented within the manuscript and the Appendix A.

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
