# Peer review of "Transcriptome Analysis Reveals Cross-Talk between the Flagellar Transcriptional Hierarchy and Secretion System in Plesiomonas shigelloides"

_ijms, 2024, doi:10.3390/ijms25137375_

Round 1

Reviewer 1 Report

Comments and Suggestions for Authors

The authors examine the transcription hierarchy involved in polar and lateral flagellar gene regulation in Pleisomonas shigelloides by deleting genes known to be involved in regulating transcription of flagellar genes in other bacteria and characterizing the resulting mutant. Genes that were deleted included flaK, flaM, fliA, and fliAL. Transcriptomes of the mutant strains were analyzed by RNA-Seq, and the results of the RNA-Seq data for select genes were validated by qRT-PCR and lux reporter genes. In addition, the authors show that FlaK and FlaM bind to the promoter regions of some of their predicted target genes in gel shift assays. The authors report that genes encoding type VI secretion system (T6SS) and type II secretion system (T2SS-2) genes were down-regulated in the flaK, flaM, fliA, and fliAL mutants, suggesting that there is cross-talk between the flagellar gene transcription hierarchy and the secretion systems. T6SS and T2SS-2 are used to kill neighboring bacterial cells. In support of the results of the RNA-seq assays, flaK, flaM, fliA, and fliAL mutants were reduced in their ability to kill Escherichia coli cells compared to wild type. Finally, the authors report that loss of the flagellar gene regulators alters the expression of some metabolic genes, including genes involved in amino acid biosynthesis, hydrogen metabolism, central metabolism, and anaerobic respiration.

Major comments:

1. It is not clear if the apparent changes in expression of genes related to metabolism in the strains where the flagellar regulatory genes were deleted are physiologically relevant. Demonstrating that the flaK, flaM, fliA, or fliAL mutants have altered growth phenotypes would help validate the physiological relevance of the observed changes in gene expression in the mutants. For example, the authors report that genes involved in amino acid biosynthesis were down-regulated in the flaM mutant. One might expect the flaM mutant to have a growth defect in media that lacked these amino acids. The authors did examine the growth rates of the mutants, but this was done in rich media (LB and DMEM) where one would not expect to observe any differences in growth rate between the mutants and wild type. Growth of the mutants should be examined in a defined minimal medium that does not contain amino acids to see if any of the mutants exhibit any growth defects. Alternatively, the authors could determine if the mutants have reduced hydrogenase or nitrate reductase activities since genes involved in the processes were reported to be down-regulated in some of the mutants.

2. The amount of data presented in the figures is overwhelming, which makes it difficult to follow. There are 16 panels within Figure 1, 22 panels within Figure 2, and 9 panels within Figure 3, which is excessive. The authors should move some of the panels in the figures to separate figures or to Supplemental Information.

Minor comments:

1. FlaK is first mentioned in the Introduction on line 75. The authors indicate that LafK many compensate for FlaK’s role in enabling swimming. FlaM is not mentioned in the Introduction until the last paragraph, and its function is not indicated. The functions of FlaK and FlaM should be discussed in the Introduction since the characterization of these proteins are significant part of the manuscript.

2. What was the rationale for picking the 3 up-regulated and 3 down-regulated DEGs for validation by qRT-PCR? What are the functions of these genes?

3. Figures S2 and S3 are screen shots from blastp analyses, which are not very informative.

Comments on the Quality of English Language

Line 189 – change “FlaK almost positively regulated all polar flagellar gene clusters” to “FlaK positively regulated nearly all of the polar flagellar gene clusters”.

Author Response

Major Comments:

  1. 1.It is not clear if the apparent changes in expression of genes related to metabolism in the strains where the flagellar regulatory genes were deleted are physiologically relevant. Demonstrating that the flaKflaMfliA, or fliAmutants have altered growth phenotypes would help validate the physiological relevance of the observed changes in gene expression in the mutants. For example, the authors report that genes involved in amino acid biosynthesis were down-regulated in the flaM One might expect the flaMmutant to have a growth defect in media that lacked these amino acids. The authors did examine the growth rates of the mutants, but this was done in rich media (LB and DMEM) where one would not expect to observe any differences in growth rate between the mutants and wild type. Growth of the mutants should be examined in a defined minimal medium that does not contain amino acids to see if any of the mutants exhibit any growth defects. Alternatively, the authors could determine if the mutants have reduced hydrogenase or nitrate reductase activities since genes involved in the processes were reported to be down-regulated in some of the mutants. 

Our response: Thank you for the suggestions. In this study, we primarily wanted to confirm the effect of flaK, flaM, fliA, or fliAL mutants in LB and DMEM on the overall growth rate of P. shigelloides compared to WT; thus, we carried out some molecular biology investigations and phenotype experiments. Because each deletion strain can influence several physiological metabolic gene expressions, we will accept your suggestion to verify the growth of mutants in the corresponding basic medium in subsequent experiments.

  1. 2.The amount of data presented in the figures is overwhelming, which makes it difficult to follow. There are 16 panels within Figure 1, 22 panels within Figure 2, and 9 panels within Figure 3, which is excessive. The authors should move some of the panels in the figures to separate figures or to Supplemental Information.

Our response: Thanks for your suggestion. First of all, we apologize for the amount of data involved in the results affecting your reading experience. After consideration, we believe that the figures currently in the manuscript should be retained, which will facilitate our presentation of the completeness and logic of the results. In addition, we hope that the legend below the figures will help you understand.

Minor Comments:

  1. FlaK is first mentioned in the Introduction on line 75. The authors indicate that LafK many compensate for FlaK’s role in enabling swimming. FlaM is not mentioned in the Introduction until the last paragraph, and its function is not indicated. The functions of FlaK and FlaM should be discussed in the Introduction since the characterization of these proteins are significant part of the manuscript.

Our response: Thanks for your suggestion. The relevant explanation was added in page 3, lines 144-149 of the revised manuscript.

  1. What was the rationale for picking the 3 up-regulated and 3 down-regulated DEGs for validation by qRT-PCR? What are the functions of these genes?

Our response: Thanks for your suggestion. We selected 3 up-regulated and 3 down-regulated DEGs in RNA-seq except flagella genes, T2SS-2 genes, and T6SS genes to further verify the validity of RNA-seq. Those selected DEGs are involved in myo-inositol catabolism (iolC/iolB/iolE), starch and sucrose metabolism (treB/treC), citrate cycle (aceE/aceF/aceA), hydrogenase (hybC/hybB/hybA), periplasmic nitrate reductase (napF/napA/napG/napB/napC), Fe-S cluster assembly transcriptional regulator (iscR), and bifunctional UDP-N-acetylglucosamine diphosphorylase/glucosamine-1-phosphate N-acetyltransferase (glmu).

  1. Figures S2 and S3 are screen shots from blastp analyses, which are not very informative.

Our response: Thanks for your suggestion. Please refer to the newly uploaded and revised supplemental file for more details.

  1. Comments on the Quality of English Language. Line 189–change “FlaK almost positively regulated all polar flagellar gene clusters” to “FlaK positively regulated nearly all of the polar flagellar gene clusters”.

Our response: Thanks for your suggestion. Modified. The relevant explanation was added in page 8, lines 204-205 of the revised manuscript.

Reviewer 2 Report

Comments and Suggestions for Authors

Dear Authors,

I am writing to provide my review of the manuscript titled “Transcriptome analysis reveals cross-talk between the flagellar transcriptional hierarchy and secretion system in Plesiomonas shigelloides” which has been submitted for publication in IJMS  Journal. 

This study aimed to uncover the unknown flagellar transcriptional hierarchy of Plesiomonas shigelloides, a unique gram-negative bacillus capable of producing both polar and lateral flagella and causing various human illnesses.

Overall, this study provides foundational insights into the flagellar transcriptional hierarchy of P. shigelloides and its interplay with motility, virulence, and metabolic processes.

During the review, I highlighted the following critical points.

Abstract:

1.     Plesiomonas shigelloides does not belong to the Enterobacteriaceae family but to the Vibrionaceae family. This could be a significant error that compromises the credibility of the entire study.

2.     The abstract references specific techniques (qRT-PCR, EMSA, promoter-lux fusions) without explaining what they are or how they were used in the study. This could make it difficult for non-specialist readers to understand.

3.     There is not enough context provided to understand the significance of the results related to flagellar regulation and the secretion of the T6SS and T2SS-2 systems. The abstract mentions the reduced infection and killing capacity without discussing the biological or clinical implications of these results.

4.     The conclusion of the abstract is rather generic and does not provide a clear indication of how the study's results can be applied or what the next steps in the research are.

Introduction

  1. The introduction is extremely long and detailed, including a lot of information that could be distributed into other sections of the document, such as the literature review. This makes it difficult to maintain attention and follow the logical thread.
  2. Many details are provided about the regulatory mechanisms of flagella in other bacteria, which can distract from the main focus of the study on P. shigelloides. The relevance of this information to the study in question is not always clear.
  3. Some concepts are repeated multiple times, such as the importance of motility for bacterial virulence. This repetition can be redundant and reduce the effectiveness of the text's communication.
  4. Although the introduction begins with a description of P. shigelloides, much of the text focuses on other bacteria. It would be more useful to focus the introduction more on the specifics of P. shigelloides and the existing knowledge gaps regarding this bacterium.

Results:

1.     The text is very long and detailed, making it difficult to follow the logical thread and identify the key points. A clearer and more concise summary of the main results would help readers better understand the data presented.

2.     Although techniques such as RNA-Seq, qRT-PCR, and EMSA are mentioned, sufficient details on the protocols used are not provided. The lack of methodological descriptions can compromise the reproducibility of the study.

3.     Although the results are detailed, there is a lack of in-depth discussion on the biological implications of the findings, particularly regarding the virulence and pathogenicity of P. shigelloides. A more comprehensive discussion would help contextualize the importance of the results.

Discussion:

  1. Many points are repeated from the previous sections (Abstract, Introduction, and Results), such as the description of the regulators FlaK, FlaM, FliA, and FliAL. This redundancy makes the discussion lengthy and less effective.
  2. The discussion attempts to connect the results with previous studies but often fails to clearly explain how these results integrate or differ from those prior findings. For example, the connection between the T6SS and T2SS-2 secretion systems and the flagellar system is not adequately explored.
  3. Some statements, such as the hypothesis that FlaK could replace LafK in the regulation of the lateral flagellar system, are speculative and not supported by sufficient data. The need for further experimental confirmation is mentioned but not adequately addressed.
  4. Some sentences are ambiguous or inconsistent. For example, the sentence "While the aim of this study was to reveal the unreported flagellar hierarchy regulatory system..." is incomplete and seems out of context.
  5. The discussion covers a wide range of topics, from flagellar regulation to secretion systems and metabolic pathways, without a clear and central focus. This makes it difficult for readers to understand the main message of the study.

Materials and methods

1.     The description of bacterial strains and plasmids is limited to mentioning a table (Table S1). It would be helpful to include a brief description of the strains and plasmids used in the text to make the section more complete and understandable without necessarily having to refer to the table.

2.     Incomplete description of growth conditions: Although different types of media and growth conditions are mentioned, the conditions for all experimental stages are not specified. For example, it is unclear if all described conditions apply to all experiments or if there are specific variations for each type of experiment.

3.     The genetic manipulation method is described very briefly and refers to a previous source. A more detailed and specific description of the protocol, including any controls and optimizations, would improve transparency and reproducibility.

4.     The section on RNA isolation and transcriptome sequencing is generic. Critical details are missing, such as the number of biological and technical replicates, RNA quality controls (beyond mentioning purity measurement), and the specifics of bioinformatic analyses conducted on the sequencing data.

5.     In the qRT-PCR section, details regarding reaction conditions, negative and positive controls used, and strategies to minimize technical variations are not described.

6.     In the description of the motility assay and transmission electron microscopy (TEM), details on the culture conditions of the agar plates and specific parameters for the preparation and visualization of the cells under TEM are missing.

Comments on the Quality of English Language

Moderate editing of English language required

Author Response

Major Comments:

Abstract:

  1. 1.Plesiomonas shigelloidesdoes not belong to the Enterobacteriaceae family but to the Vibrionaceae family. This could be a significant error that compromises the credibility of the entire study.

Our response: Thanks for your advice. After many years in the family Vibrionaceae, the genus Plesiomonas, represented by a single species, Plesiomonas shigelloides, currently resides in the family Enterobacteriaceae, although its most appropriate phylogenetic position may yet to be determined [1].

  • Janda, J. M., Abbott, S. L., & McIver, C. J. (2016). Plesiomonas shigelloidesClinical microbiology reviews, 29(2), 349–374. https://doi.org/10.1128/CMR.00103-15

  1. 2.The abstract references specific techniques (qRT-PCR, EMSA, promoter-lux fusions) without explaining what they are or how they were used in the study. This could make it difficult for non-specialist readers to understand.

Our response: Thank you for your attention. We apologize for any inconvenience caused by the use of the abbreviations of method techniques in the abstract, and we have provided the complete name of each abbreviation in the revised manuscript. Additionally, the Materials and methods section of the manuscript describes the procedures and principles of the methods and techniques used in this study; we have added more details in the revised manuscript based on your suggestions.

  1. 3.There is not enough context provided to understand the significance of the results related to flagellar regulation and the secretion of the T6SS and T2SS-2 systems. The abstract mentions the reduced infection and killing capacity without discussing the biological or clinical implications of these results.

Our response: Thanks for your suggestion. This work primarily used RNA-seq and qRT-PCR to confirm the down-regulation of T2SS-2 and T6SS genes due to the deletion of four flagella regulators (FlaK, FlaM, FliA, and FliAL), which are in charge of polar and lateral flagellar regulation in P. shigelloides. Furthermore, as demonstrated by the killing assays, the four gene deletion strains had a lower capacity for killing when compared to WT. These findings suggest a connection between the flagellar system and the secretion systems, T6SS and T2SS-2; further research will ascertain the precise mechanism. Additionally, the biological or clinical implications were added to the abstract of the revised manuscript.

  1. 4.The conclusion of the abstract is rather generic and does not provide a clear indication of how the study's results can be applied or what the next steps in the research are.

Our response: Thanks for your suggestion. The relevant explanation was added in page 1, lines 34-37 of the revised manuscript.

Introduction:

  1. The introduction is extremely long and detailed, including a lot of information that could be distributed into other sections of the document, such as the literature review. This makes it difficult to maintain attention and follow the logical thread.

Our response: Thank you for your suggestion. We have made adjustments to the introduction section based on your recommendations.

  1. Many details are provided about the regulatory mechanisms of flagella in other bacteria, which can distract from the main focus of the study on P. shigelloides. The relevance of this information to the study in question is not always clear.

Our response: Thank you for your suggestion. We have simplified the regulatory mechanisms of flagella in other bacteria in the introduction section of the revised manuscript. The relevant explanation was added in page 2, lines 52-84 of the revised manuscript.

  1. Some concepts are repeated multiple times, such as the importance of motility for bacterial virulence. This repetition can be redundant and reduce the effectiveness of the text's communication.

Our response: Thank you for your suggestion. We have simplified the importance of motility for bacterial virulence in the introduction section of the revised manuscript. The relevant explanation was added in page 2-3, lines 85-123 of the revised manuscript.

  1. Although the introduction begins with a description of P. shigelloides, much of the text focuses on other bacteria. It would be more useful to focus the introduction more on the specifics of P. shigelloidesand the existing knowledge gaps regarding this bacterium.

Our response: Thank you for your suggestion. Currently, there are few studies about P. shigelloides' flagella motility, virulence, or other physiological metabolic control systems, and we cite as much as possible, which is why we are committed to studying P. shigelloides and gradually filling the research gap.

Resutls:

  1. The text is very long and detailed, making it difficult to follow the logical thread and identify the key points. A clearer and more concise summary of the main results would help readers better understand the data presented.

Our response: Thank you for your suggestion. We have simplified the main results in the revised manuscript.

  1. Although techniques such as RNA-Seq, qRT-PCR, and EMSA are mentioned, sufficient details on the protocols used are not provided. The lack of methodological descriptions can compromise the reproducibility of the study.

Our response: Thank you for your suggestion. The Materials and methods section of the manuscript describes the procedures and principles of the methods and techniques used in this study; we have added more details in the revised manuscript based on your suggestions.

  1. Although the results are detailed, there is a lack of in-depth discussion on the biological implications of the findings, particularly regarding the virulence and pathogenicity of P. shigelloides. A more comprehensive discussion would help contextualize the importance of the results.

Our response: Thank you for your suggestion. The discussion on the biological implications of the findings has been added to page 9, lines 260-263 and page 16, lines 314-319 of the revised manuscript.

Discussion:

  1. Many points are repeated from the previous sections (Abstract, Introduction, and Results), such as the description of the regulators FlaK, FlaM, FliA, and FliAL. This redundancy makes the discussion lengthy and less effective.

Our response: Thank you for your suggestion. We have simplified the description of the regulators FlaK, FlaM, FliA, and FliAL in the discussion section of the revised manuscript.

  1. The discussion attempts to connect the results with previous studies but often fails to clearly explain how these results integrate or differ from those prior findings. For example, the connection between the T6SS and T2SS-2 secretion systems and the flagellar system is not adequately explored.

Our response: Thank you for your suggestion. In the third paragraph of the discussion section, we discussed the cross-talk between the flagellar system and T6SS in other bacteria, but the flagellar gene that influences the T6SS system is distinct. In this study, we confirmed that FlaK, FlaM, and FliAL were positively correlated with T6SS gene expression, while FlaM and FliA were positively correlated with T2SS-2 gene expression by using RNA-seq and qRT-PCR. Furthermore, as demonstrated by the killing assays, the four gene deletion strains had a lower capacity for killing when compared to WT. These findings suggest a connection between the flagellar system and the secretion systems, T6SS and T2SS-2; further research will ascertain the precise mechanism.

  1. Some statements, such as the hypothesis that FlaK could replace LafK in the regulation of the lateral flagellar system, are speculative and not supported by sufficient data. The need for further experimental confirmation is mentioned but not adequately addressed.

Our response: Thank you for your suggestion. Similar to your viewpoint, we show this hypothesis that FlaK might replace LafK's function in regulating the lateral flagellar system in both the results section and the discussion section. We also do not deny the possibility that the lateral flagellar system is regulated by unidentified transcriptional regulators. In the next study, we will continue to verify this fact with more evidence.

  1. Some sentences are ambiguous or inconsistent. For example, the sentence "While the aim of this study was to reveal the unreported flagellar hierarchy regulatory system..." is incomplete and seems out of context.

Our response: Thank you for your suggestion. After careful consideration, we removed this sentence without affecting the previous or subsequent statements of content.

  1. The discussion covers a wide range of topics, from flagellar regulation to secretion systems and metabolic pathways, without a clear and central focus. This makes it difficult for readers to understand the main message of the study.

Our response: Thank you for your suggestion. The discussion section focuses on three aspects: the transcriptional hierarchy of flagella, flagella's association with bacterial pathogenicity, and flagella's association with bacterial physiology and metabolism. Since we primarily used RNA-seq to examine the effects of four flagellar regulators on motility, the secretion system, and physiological metabolism, these three sections correspond to our study's content and results.

Materials and methods:

  1. The description of bacterial strains and plasmids is limited to mentioning a table (Table S1). It would be helpful to include a brief description of the strains and plasmids used in the text to make the section more complete and understandable without necessarily having to refer to the table.

Our response: Thank you for your suggestion. We mentioned the strains and plasmids mainly used in this study in the Materials and Methods sections, such as P. shigelloides, E. coli S17-1 λpir, gene deletion strains, complementation strains, E. coli MG1655, pRE112, and pBAD33, etc.

  1. 2.Incomplete description of growth conditions: Although different types of media and growth conditions are mentioned, the conditions for all experimental stages are not specified. For example, it is unclear if all described conditions apply to all experiments or if there are specific variations for each type of experiment.

Our response: Thank you for your suggestion. We have completed growth temperature and growth conditions in each of the methods.

  1. 3.The genetic manipulation method is described very briefly and refers to a previous source. A more detailed and specific description of the protocol, including any controls and optimizations, would improve transparency and reproducibility.

Our response: Thank you for your suggestion. The details and procedures of the genetic manipulation method have been refined and supplemented in the revised manuscript. The relevant explanation was added in page 23, lines 471-480 of the revised manuscript.

  1. 4.The section on RNA isolation and transcriptome sequencing is generic. Critical details are missing, such as the number of biological and technical replicates, RNA quality controls (beyond mentioning purity measurement), and the specifics of bioinformatic analyses conducted on the sequencing data.

Our response: Thank you for your suggestion. We have supplemented the details of the number of biological and RNA quality controls in the revised manuscript. However, the specifics of bioinformatic studies performed on sequencing data contain a lot of information, so we believe it is inappropriate to include them in the material and technique section. After careful deliberation, we decided to include it here for your reference.

The RNA-seq data analysis includes the following contents and details:

  1. Quality control

The image data measured by the high-throughput sequencer are converted into sequence data (reads) by CASAVA base recognition. Raw data (raw reads) of fastq format were firstly processed through in-house perl scripts. In this step, clean data (clean reads) were obtained by removing reads containing adapter, reads containing N base and low quality reads from raw data. At the same time, Q20, Q30 and GC content the clean data were calculated. All the downstream analyses were based on the clean data with high quality.

  1. 2. Reads mapping to the reference genome

Reference genome and gene model annotation files were downloaded from genome website directly. Both building index of reference genome and aligning clean reads to reference genome were used Bowtie2 (2.3.4.3).

  1. 3. Novel gene and gene structure analysing

Rockhopper (1.2.1) software was used to identify novel genes, operon, TSS, TTS and Cis-natural antisense transcripts. It can be used for efficient and accurate analysis of bacterial RNA-seq data, and that it can aid with elucidation of bacterial transcriptomes. Then, we extract upstream 700bp sequence of Transcription Start Site for predicting promoter using TDNN(Time-DelayNeural Network).

  1. 4. Predict UTR

According to the information of Transcription Start Site(Transcription terminal Site) and Translation start site(Translation terminal site), we extracted 5’UTR(3’UTR) sequences. Then, RBSfinder (v1.0) and TransTermH (v2.0.9) were used to predict SD sequence and terminator sequence respectively.

  1. 5. Analysis of sRNA

Rockhopper was used to identify new intergenic region transcripts, and Blastx was compared with the nr library to annotate the newly predicted transgenic regions, and the unmarked transcripts were used as candidate non-coding sRNAs. RNA fold (1.8.5) and Inta RNA (1.8.5) were used to predict secondary structure and target gene respectively.

  1. 6. Quantification of gene expression level

HTSeq (v0.9.1) was used to count the reads numbers mapped to each gene. And then FPKM of each gene was calculated based on the length of the gene and reads count mapped to this gene. FPKM, expected number of Fragments Per Kilobase of transcript sequence per Millions base pairs sequenced, considers the effect of sequencing depth and gene length for the reads count at the same time, and is currently the most commonly used method for estimating gene expression levels.

  1. 7. Differential expression analysis

Differential expression analysis of two conditions/groups (two biological replicates per condition) was performed using the DESeq2 R package (1.20.0). DESeq2 provide statistical routines for determining differential expression in digital gene expression data using a model based on the negative binomial distribution. The resulting P-values were adjusted using the Benjamini and Hochberg’s approach for controlling the false discovery rate. The criteria for differentially expressed genes (DEGs) were set as |log2 fold change| ³1 and adjusted P-value (padj) £ 0.05. 

  1. 8. GO and KEGG enrichment analysis of differentially expressed genes

Gene Ontology (GO) enrichment analysis of differentially expressed genes was implemented by the clusterProfiler R package (3.8.1), in which gene length bias was corrected. GO terms with corrected Pvalue less than 0.05 were considered significantly enriched by differential expressed genes. KEGG is a database resource for understanding high-level functions and utilities of the biological system, such as the cell, the organism and the ecosystem, from molecular-level information, especially large-scale molecular datasets generated by genome sequencing and other high-through put experimental technologies (http://www.genome.jp/kegg/). We used clusterProfiler R package to test the statistical enrichment of differential expression genes in KEGG pathways.

  1. 9. Gene Set Enrichment Analysis

Gene Set Enrichment Analysis (GSEA) is a computational approach to determine if a pre-defined Gene Set can show a significant consistent difference between two biological states. The genes were ranked according to the degree of differential expression in the two samples, and then the predefined Gene Set were tested to see if they were enriched at the top or bottom of the list. Gene set enrichment analysis can include subtle expression changes. We use the local version of the GSEA analysis tool http://www.broadinstitute.org/gsea/index.jsp, GO, KEGG data set were used for GSEA independently.

  1. 10. SNP analysis

Picard-tools (v1.96) and samtools (v0.1.18) were used to sort, mark duplicated reads and reorder the bam alignment results of each sample. GATK2 (v3.5) software was used to perform SNP calling.

  1. 5.In the qRT-PCR section, details regarding reaction conditions, negative and positive controls used, and strategies to minimize technical variations are not described.

Our response: Thank you for your suggestion. We have supplemented the details of the qRT-PCR section in the revised manuscript. The relevant explanation was added in page 24, lines 517-526 of the revised manuscript.

  1. 6.In the description of the motility assay and transmission electron microscopy (TEM), details on the culture conditions of the agar plates and specific parameters for the preparation and visualization of the cells under TEM are missing.

Our response: Thank you for your suggestion. We have supplemented the description of the motility assay and transmission electron microscopy (TEM) in the revised manuscript. The relevant explanation was added in page 24, lines 528-544 of the revised manuscript.

Round 2

Reviewer 1 Report

Comments and Suggestions for Authors

The authors have adequately addressed my comment regarding the growth rates of the mutants. The authors made several changes in the text based on the reviewers' comments. The introduced changes improve the text, but some of the additions are poorly written. I suggest the following changes for the revised sections.

Line 33 – Change to “Overall, this study aims to reveal the transcriptional hierarchy that controls flagellar gene expression in P. shigelloides, as well as ….”

Line 117 – Change to “Merino and co-workers reported that although P. shigelloides lacks a FlrB ortholog, P. shigelloides FlrC (FlaM) contains the PAS and His Kinase A domains found in FlrC proteins of Vibrio species and A. hydrophila, indicating that P . shigelloides FlaM may activate transcription from s54-dependent promoters of class III genes [39].”

Lines 266-271 – Change to “Taken together, our findings unveil a connection between the flagellar and secretory systems in P. shigelloides and provide new insights into the pathogenesis of this bacterium.”

Some of the panels in Figures 1 and 2 are cut off.

The authors do no appreciate the burden that the extensive number of panels in the figures places on the reader. Figure 1 takes up 4 pages and Figure 2 takes up 7 pages. This number of pages for figures makes it difficult to follow as one reads the text and examines the figures. I appreciate the amount of work that the authors have done, but the information in the figures can be presented in a more organized fashion. One potential way of organizing the information would be to break up the figures to create separate figures for each of the mutants.

Comments on the Quality of English Language

Problems on the quality of English were noted in the Comments section above.

Author Response

  1. 1.Line 33 – Change to “Overall, this study aims to reveal the transcriptional hierarchy that controls flagellar gene expression in  shigelloides, as well as ….”

Our response: Thank you very much for your suggestions. Changed.

  1. 2.Line 117 – Change to “Merino and co-workers reported that although P. shigelloides lacks a FlrB ortholog, P. shigelloides FlrC (FlaM) contains the PAS and His Kinase A domains found in FlrC proteins of Vibrio species and A. hydrophila, indicating that P . shigelloides FlaM may activate transcription from s54-dependent promoters of class III genes [39].”

Our response: Thank you very much for your suggestions. Changed. 

  1. Lines 266-271 – Change to “Taken together, our findings unveil a connection between the flagellar and secretory systems in P. shigelloides and provide new insights into the pathogenesis of this bacterium.”

Our response: Thank you very much for your suggestions. Changed.

  1. Some of the panels in Figures 1 and 2 are cut off.

Our response: Thank you very much for your attention. Modified.

  1. The authors do no appreciate the burden that the extensive number of panels in the figures places on the reader. Figure 1 takes up 4 pages and Figure 2 takes up 7 pages. This number of pages for figures makes it difficult to follow as one reads the text and examines the figures. I appreciate the amount of work that the authors have done, but the information in the figures can be presented in a more organized fashion. One potential way of organizing the information would be to break up the figures to create separate figures for each of the mutants.

Our response: Thank you very much for your suggestions. Based on your suggestions, we moved some of the resulting images from Figure 1 to the supplemental file and rearranged the photos from Figure 2.
